# Chiral Modulations in Non-Heisenberg Models of Non-Centrosymmetric Magnets Near the Ordering Temperatures

Andrey O. Leonov [1,2,3]

1    International Institute for Sustainability with Knotted Chiral Meta Matter, Higashihiroshima 739-8511, Japan; leonov@hiroshima-u.ac.jp
2    Department of Chemistry, Faculty of Science, Hiroshima University, Higashihiroshima 739-8526, Japan
3    IFW Dresden, Postfach 270016, D-01171 Dresden, Germany

**Abstract:** The structure of skyrmion and spiral solutions, investigated within the phenomenological Dzyaloshinskii model of chiral magnets near the ordering temperatures, is characterized by the strong interplay between longitudinal and angular order parameters, which may be responsible for experimentally observed precursor effects. Within the precursor regions, additional effects, such as pressure, electric fields, chemical doping, uniaxial strains and/or magnetocrystalline anisotropies, modify the energetic landscape and may even lead to the stability of such exotic phases as a square staggered lattice of half-skyrmions, the internal structure of which employs the concept of the "soft" modulus and contains points with zero modulus value. Here, we additionally alter the stiffness of the magnetization modulus to favor one- and two-dimensional modulated states with large modulations of the order parameter magnitude. The computed phase diagram, which omits any additional effects, exhibits stability pockets with a square half-skyrmion lattice, a hexagonal skyrmion lattice with the magnetization in the center of the cells parallel to the applied magnetic field, and helicoids with propagation transverse to the field, i.e., those phases in which the notion of localized defects is replaced by the picture of a smooth but more complex tiling of space. We note that the results can be adapted to metallic glasses, in which the energy contributions are the same and originate from the inherent frustration in the models, and chiral liquid crystals with a different ratio of elastic constants.

**Keywords:** skyrmion; chiral magnets; Dzyaloshinskii–Moriya interaction; MnSi; FeGe

## 1. Introduction

In non-centrosymmetric magnetic systems, chiral Dzyaloshinskii–Moriya interactions (DMIs) based on the relativistic spin–orbit couplings [1,2] provide a unique mechanism for the stabilization of solitonic textures in two dimensions—baby skyrmions—which are extended into the third direction as skyrmion strings [3–5]. These skyrmions may exist as localized particle-like countable excitations of the homogeneously magnetized state [6,7] with relevant length scale tuned by the competition between direct exchange and DMI [8,9]. Alternatively, skyrmions may condense into multiply modulated phases—skyrmion lattices [10,11]. Whereas the long-period one-dimensional spiral modulations with a fixed sense of rotation due to DMI have been known for a long time [1,12,13], skyrmionic textures were found to form ground states in the cubic helimagnets like MnSi [14] or FeGe [15] near the ordering temperature $T_C$ only in 2009 [16]. In the applied magnetic field $H$, the skyrmions are responsible for the appearance of a small closed area in the $(H, T)$-phase diagram—A-phase [16,17]. Within a small A-pocket, the skyrmion lattice (SkL) appears spontaneously, and its stability is commonly attributed to the thermal fluctuations, which become sizable at relatively high temperatures [16,18]. At the same time, the boundaries of the A-phase can be drastically changed by applying pressure [19], electric fields [20–22],

chemical doping [23], or uniaxial strains [24,25]. Weak magnetocrystalline anisotropies may also modify the energetic landscape and eventually favor the SkL in the A-phase pocket [26,27]. In zero field, the skyrmions may underlie a spin liquid phase between the low-temperature helical and the high-temperature paramagnetic state [28].

From recent numerical studies within the phenomenological (Dzyaloshinskii) theory [29,30], it is known that these "high-temperature" skyrmions, underlying described precursor phenomena, possess a number of unique properties: (i) the interaction between the chiral skyrmions, being repulsive in a broad temperature range, changes into attraction at high temperatures [29], and (ii) this leads to a remarkable confinement effect—near the ordering temperatures, skyrmions exist only as bound states, and different skyrmion mesophases as square half- or $\pm\pi$-skyrmion lattices are formed by an unusual instability-type nucleation transition [30]. In $\pm\pi$-SkLs, the magnetization in the center of the lattice cell is anti-parallel and parallel to the applied field. The half-SkL is formed by cells with up and down magnetization in the center and in-plane magnetization along the cell boundaries. Such features of confined skyrmions distinguish them from their "low-temperature" counterparts visualized by Lorentz microscopy, e.g., in nanolayers of FeGe [31] and $Fe_{0.5}Co_{0.5}Si$ [32] far from $T_C$ and theoretically studied in early works by A. Bogdanov and co-workers [3,6].

In the present paper, we suggest another mechanism of skyrmion stability near the ordering temperatures achieved within the framework of the modified Dzyaloshinskii model for metallic cubic helimagnets [11]. We neglect different secondary effects and provide a more realistic description of the inhomogeneous twisted magnetic structure in these mesophases. We show that confined chiral modulations are very sensitive to values of the stiffness parameter $\eta$ characterizing a modulus field. The magnetic phase diagram calculated for $\eta = 0.8$ exhibits pockets with square half-SkLs, hexagonal SkLs with the magnetization in the center of the cells parallel to the applied field, and helicoids with propagation transverse to the applied field. Therefore, the present results change the picture of the formation and evolution of chiral modulated textures and shed new light on the problem of precursor states observed as blue phases in chiral nematics [33] and in chiral magnets [16,28].

## 2. Phenomenological Theory and Equations

The equilibrium chiral modulations in cubic helimagnets near the ordering temperatures are derived by the minimization of the phenomenological magnetic energy [1,12]:

$$W = \int d^3r [A(\text{grad}\mathbf{M})^2 + D\mathbf{M} \cdot \text{rot}\mathbf{M} - \mathbf{H} \cdot \mathbf{M} + a_1 M^2 + a_2 M^4] \tag{1}$$

where the first and the second terms with coefficients $A$ and $D$ are isotropic and antisymmetric exchange interactions; the third term is Zeeman energy density with $\mathbf{H}$ being an applied magnetic field; and the fourth and fifth terms represent the Landau expansion near the ordering temperature. These terms reflect the phenomenon that near the ordering temperatures, the magnetization amplitude varies under the influence of the applied magnetic field and temperature. According to Landau, this process is described by supplementing the magnetic energy with the additional homogeneous free energy term with even powers of the magnetization. Although higher-order terms can be included, it is a reasonable approximation to consider the series to fourth order in the order parameter as long as the order parameter is small.

Functional (1) includes all necessary (primary) interactions *essential* to stabilize the skyrmion and helical phases.

By rescaling the spatial variable in (1), $\mathbf{x} = \mathbf{r}/L_D$, the magnetic field $\mathbf{h} = \mathbf{H}/H_0$, and the magnetization $\mathbf{m} = \mathbf{M}/M_0$

$$L_D = A/D, \quad H_0 = \kappa M_0, \ M_0 = (\kappa/a_2)^{1/2}, \ \kappa = D^2/(A), \ a = a_1/\kappa = J(T - T_c)/\kappa \tag{2}$$

energy $W$ (1) can be written in the following reduced form:

$$\widetilde{W} = \int d^3x \left[ (\mathbf{grad}\, \mathbf{m})^2 - \mathbf{m} \cdot \mathrm{rot}\mathbf{m} - h(\mathbf{n} \cdot \mathbf{m}) + am^2 + m^4 \right], \tag{3}$$

where $h = |\mathbf{h}|$ and $\mathbf{n}$ is a unity vector along the applied magnetic field. The case $a_1 = 0$ corresponds to the critical temperature $a_c$, where spontaneous magnetization appears (Figure 1a).

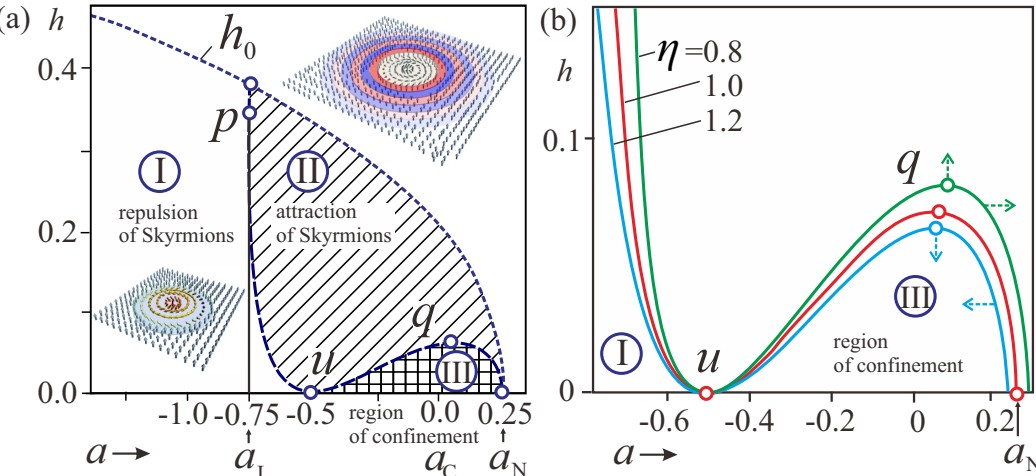

**Figure 1.** (Color online) (**a**) The diagram on plane $(a, h)$ showing the regions with different types of skyrmion–skyrmion interactions: I—repulsive interaction between isolated skyrmions; II—attractive inter-skyrmion interaction; III—the region of skyrmion confinement. In region III, skyrmions exist only as bound states since oscillations in the asymptotics of isolated skyrmions do not diminish. Above the line $h_0$, no isolated skyrmions can exist. (**b**) Critical line, which delineates different regimes, in the case of the non-Heisenberg model for different values of parameter $\eta$.

Functional (3) includes three internal variables (components of the magnetization vector $\mathbf{m}$, which contribute to the variable modulus) and two control parameters, the reduced magnetic field with amplitude $h$ and the "effective" temperature $a(T)$ (2).

### 2.1. High-Temperature Isolated Skyrmions

Isolated skyrmions can be thought of as isolated static solitonic textures localized in two spatial directions. The magnetization in the center of skyrmion pointing opposite to an applied magnetic field rotates smoothly in all radial directions and reaches the orientation along the field at the outskirt of skyrmion. The structure of isolated skyrmions near the ordering temperature is characterized by the dependence of the polar angle $\theta(\rho)$ and modulus $m(\rho)$ on the radial coordinate $\rho$ (we introduce here spherical coordinates for the magnetization $(m, \theta, \psi)$ and cylindrical coordinates for the spatial variables $(\rho, \phi, z)$) and is determined from the system of Euler equations [11,29]:

$$\begin{aligned}
& m^2\left[\theta_{\rho\rho} + \frac{\theta_\rho}{\rho} + \frac{\sin\theta\cos\theta}{\rho^2} + \frac{2\sin^2\theta}{\rho} - h\sin(\theta)\right] + 2(\theta_\rho - 1)m_\rho = 0, \\
& m_{\rho\rho} + \frac{m_\rho}{\rho} + m\left[\theta_\rho^2 + \frac{\sin^2\theta}{\rho^2} + \theta_\rho + \frac{\sin\theta\cos\theta}{\rho}\right] + 2am + 4m^3 - h\cos(\theta) = 0
\end{aligned} \tag{4}$$

with boundary conditions $\theta(0) = \pi, \theta(\infty) = 0, m(\infty) = m_0, m(0) = m_1$ and the magnetization in the homogeneous phase $m_0$ being determined from the equation $2am_0 + 4m_0^3 - h = 0$.

As it was determined in Ref. [6], the asymptotic behavior of isolated skyrmions bears exponential character: $\Delta m = (m - m_0) \propto \exp(-\alpha\rho)$, $\theta \propto \exp(-\alpha\rho)$. By substituting these into the linearized Euler Equation (4) for $\rho \to \infty$, one can find three distinct regions in the

magnetic phase diagram on the plane $(a, h)$ with different character of skyrmion–skyrmion interactions (Figure 1a): *repulsive* interactions between isolated skyrmions occur in a broad temperature range (area (I)) and are characterized by real values of parameter $\alpha \in \Re$ (I), the magnetization in such skyrmions always having the "right" rotation sense; at higher temperatures (area (II)), the skyrmion–skyrmion interaction changes to an *attractive* character with complex $\alpha \in C$ (II); finally, in area (III) near the ordering temperature, $a_N = 0.25$, only strictly confined skyrmions exist with imaginary $\alpha \in \Im$. The line separating different regimes of inter-skyrmion interaction on the plane $(a, h)$ looks like

$$h^\star = \sqrt{2 \pm P(a)}(a + 1 \pm P(a)/2), \; P(a) = \sqrt{3 + 4a} \tag{5}$$

with turning points $p$ $(-0.75, \sqrt{2}/4)$, $q$ $(0.06, 0.032\sqrt{5})$, and $u$ $(-0.5, 0)$ (dashed line in Figure 1a).

The typical solutions, such as profiles $\theta(\rho)$ and $m(\rho)$, for isolated skyrmions in each region were investigated in Refs. [29,30].As in region II, the exponents $\alpha$ are complex numbers, the skyrmion profiles display antiphased oscillations. The rotation of the magnetization in such an isolated skyrmion contains two types of rotation sense: if rotation has the "right" sense, the modulus increases, and otherwise, the modulus decreases in parts of the skyrmion with the "wrong" rotation sense. Such a unique rotational behavior of the magnetization is a consequence of the strong coupling between two order parameters in Equation (4)—modulus $m$ and angle $\theta$.

### 2.2. Energy Minimization

For rigorous minimization of the functional (3), the Euler–Lagrange equations are nonlinear partial differential equations. These equations were solved by a numerical energy minimization procedure using finite-difference discretization on grids with adjustable grid spacings and periodic boundary conditions [34]. Components of the magnetization vector **m** were evaluated in the knots of the grid, and for the calculation of the energy density (3), we used the finite-difference approximation of derivatives with different precision up to eight points as neighbors. To check the stability of the numerical routines, we additionally refined and coarsened the grids. For axial fields, we used grid spacings $\Delta_y \approx \Delta_x$ so that grids were approximately square in the $xy$ plane in order to reduce the artificial anisotropy incurred by the discretization. The final equilibrium structure for the two-dimensional modulated states was obtained according to the following iterative procedure of the energy minimization using simulated annealing and single-step Monte Carlo dynamics with the Metropolis algorithm:

(i)    The initial configuration of magnetization vectors in the grid knots for Monte Carlo annealing is chosen appropriately to ensure relaxation to a desired particle-like state.

(ii)    A point $(x_n, y_n, z_n)$ on a grid is chosen randomly. Then, the magnetization vector in that point is rotated without a change in its length. If the energy change $\Delta H_k$ associated with such a rotation is negative, the new orientation is kept.

(iii)    However, if the new state has an energy higher than the last one, it is accepted probabilistically. The probability $P$ depends upon the energy and a kinetic cycle temperature $T_k$:

$$P = \exp\left[-\frac{\Delta H_k}{k_B T_k}\right], \tag{6}$$

where $k_B$ is the Boltzmann constant. Together with probability $P$, a random number $R_k \in [0, 1]$ is generated. If $R_k < P$, the new configuration is accepted and is otherwise discarded. Generally speaking, at high temperatures $T_k$, many states will be accepted, while at low temperatures, the majority of these probabilistic moves will be rejected. Therefore, one has to choose an appropriate starting temperature for heating cycles.

(iv)    The characteristic spacings $\Delta_x$, $\Delta_y$, and $\Delta_z$ are also adjusted to promote energy relaxation. The procedure is stopped when no further reduction in energy is observed.

### 2.3. Modulated Phases Stabilized within the Model (3)

Skyrmion periodic states near the ordering temperatures, which may be "designed" with the help of the "soft-modulus" tool kit and may underlie the occurrence of an anomalous precursor regime, include the following states: (i) hexagonal lattices of $\pm\pi$-skyrmions (i.e., skyrmions of both polarities with respect to the field direction, in which the magnetization undergoes the full swing by the angle $\pi$ from the center to the outskirt of the lattice cell (Figure 2d,e); (ii) square staggered lattices of $\pi/2$-skyrmions, in which the angle is halved (Figure 2c). A square half-skyrmion lattice exploits the soft-modulus version of the magnetization to match the half-skyrmion configurations in the interstitial regions between the skyrmionic cores: then, zero magnetization points/lines replace the defects between the square plaquettes pointing up and down [11,30]. Exhaustive analysis of field- and temperature-driven evolution of skyrmion mesophases near the ordering temperature was performed in Refs. [29,30].

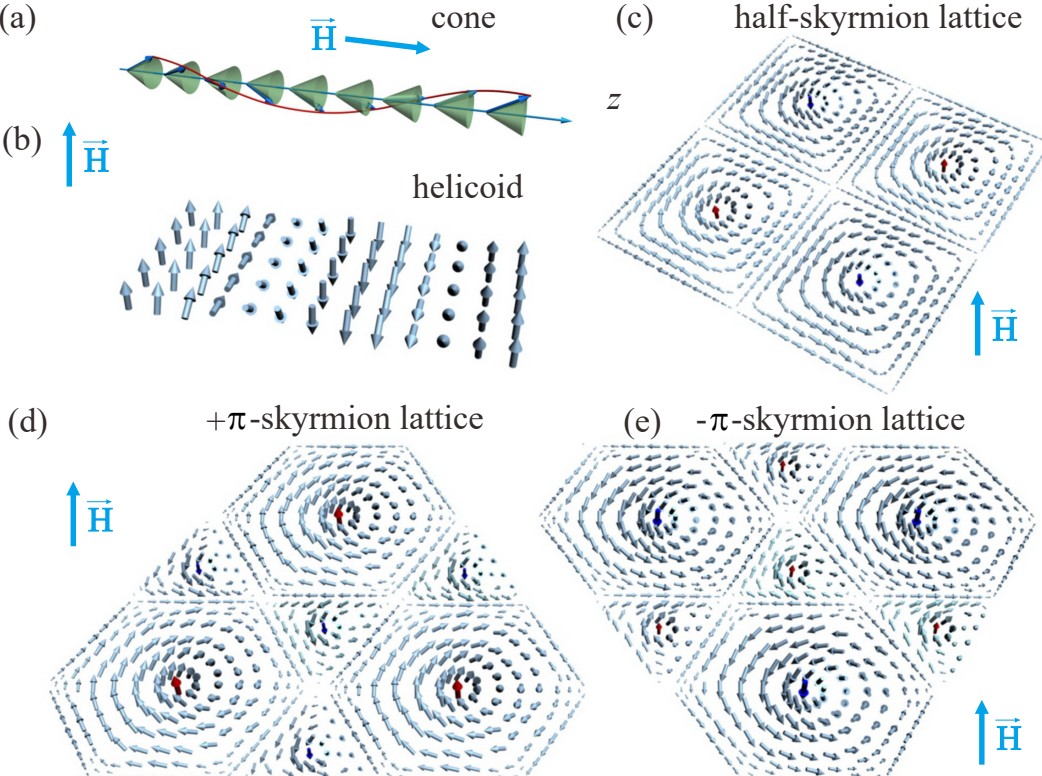

**Figure 2.** (color online) Schematics of one- and two-dimensional mesophases near the ordering temperature $T_c$, obtained as solutions of the functional (3). A conical spiral is a single-harmonic solution with the propagation direction along the magnetic field (**a**). If the propagation vector of a spiral state is perpendicular to an applied magnetic field, such a state is called helicoid (**b**). The half-skyrmion lattice (**c**) consists of cells with up (red color) and down (blue color) magnetization in the center and in-plane magnetization along the cell boundaries. In the hexagonal $+\pi$ (**d**) and $-\pi$ (**e**) skyrmion lattices, the magnetization in the center of the lattice cell is parallel and anti-parallel to the applied field, correspondingly, which is indicated by the corresponding color scheme. The cells bear unit topological charges. The central hexagon is embraced by the triangular regions.

We consider one-dimensional spiral states with their wave vectors along the field (cones, Figure 2a) or perpendicular to the field (helicoids, Figure 2b). For the isotropic model (3), the cone phase with the fixed magnetization modulus and rotation of **m** around the applied magnetic field, $\psi = z$, $\cos(\theta) = h/m$, $m = |a - 1/2|/2$, is the global energy minimum in the whole region where modulated states exist. That is why, for cubic helimagnets, the energy density (3) has to be supplemented, e.g., by anisotropic contributions,

$f_a = b_{ea} \sum_i (\partial m_i / \partial x_i)^2 + k_c \sum_i m_i^4$, where $b_{ea}$ and $k_c$ are reduced values of the exchange and cubic anisotropies [12]. These anisotropic interactions impair the ideal harmonic twisting of the cone phase and allow for the thermodynamic stability of skyrmion states as discussed in Ref. [27,30]. $Cu_2OSeO_3$ represents a unique example in the family of B20 cubic helimagnets, exhibiting two well-defined skyrmion pockets stabilized by the competing exchange and cubic anisotropies in the temperature–magnetic field phase diagram [35].

### 2.4. A Generalized Gradient Energy for a Chiral Isotropic System

A generalization of isotropic chiral magnets proposed in Ref. [11] replaces the usual Heisenberg-like exchange model (3) by a non-linear sigma model coupled to a modulus field with different stiffness $\eta$:

$$\sum_{i,j} (\partial_i m_j)^2 \rightarrow \sum_{i,j} (\partial_i m_j)^2 + (1 - \eta) \sum_{i,j} (\partial_i m)^2 \rightarrow m^2 \sum_{i,j} (\partial_i n_j)^2 + \eta \sum_i (\partial_i m)^2. \quad (7)$$

Parameter $\eta$ equals unity for a "Heisenberg" model, in chiral nematics $\eta = 1/3$ [33].

Model (7) yields a generalized gradient energy for a chiral isotropic system with a vector order parameter, which is equivalent to the phenomenological theory in the director formalism [11,33] of liquid crystals. Chiral liquid crystals are considered ideal model systems for probing the behavior of different modulated structures on the mesoscopic scale. In these systems, a surprisingly large diversity of naturally occurring and laser-generated topologically nontrivial solitons with differently knotted nematic fields has recently been investigated [36].

For $\eta > 1$, the field- and temperature-driven evolutions of skyrmion and helical states are qualitatively the same as for $\eta = 1$ [29,30]. However, for the thermodynamical stability of skyrmions, higher values of additional anisotropic contributions must be applied. The endpoints of the lines bounding the confinement region are shifted to the left (i.e., in the region of lower temperatures) with respect to $a_N = 0.25$ (blue line in Figure 1b). Therefore, the conical phase can exist for higher temperatures in comparison with skyrmions. In this case, skyrmions, being stabilized by additional energy contributions, occupy the pocket within the region of stability of the cone phase bounded by the lines of the first-order phase transition skyrmion cones.

For $\eta < 1$ on the contrary, the additional "softness" of the longitudinal order parameter makes the confined chiral modulations extremely sensitive to the applied magnetic field, temperature, and anisotropic energy contributions: different chiral states undergo a very complex sequence of phase transitions. In a zero magnetic field, the region of confinement extends to the temperatures higher than $a_N$ (green line in Figure 1b). This means that skyrmions and helicoids can exist and compete for the thermodynamical stability for $a > a_N$. Cones appear only for $a \leq a_N$, independent of the value of $\eta$. Such a phase offset becomes crucial even for a negligible difference in stiffness coefficients.

The phase diagram of states for model (3), including a generalized gradient energy with $\eta = 0.8$, is plotted in Figure 3a.

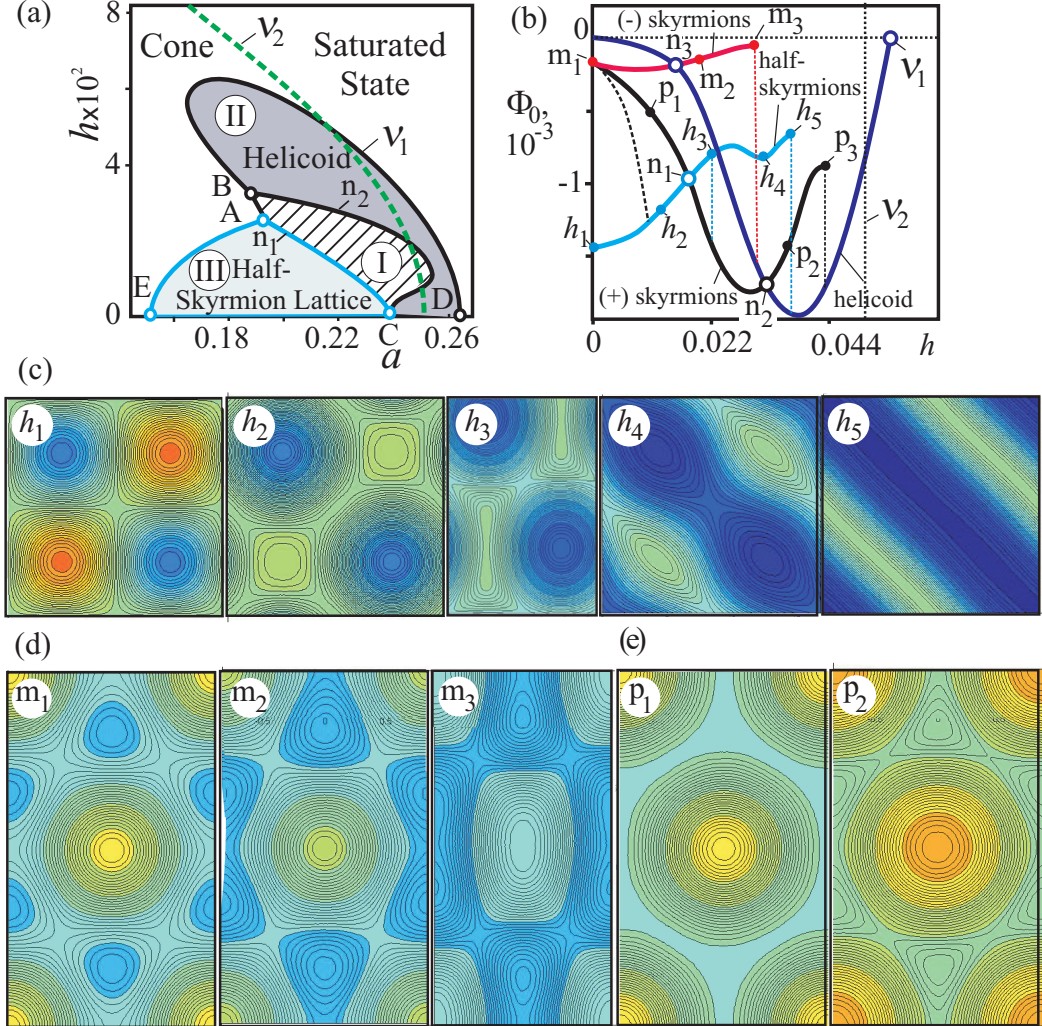

**Figure 3.** (Color online) (**a**) Theoretical phase diagram for chiral magnets near magnetic ordering according to the modified non-linear sigma model [11]. In larger applied fields, i.e., in the A-region, a densely packed full skyrmion lattice is found in region (I). The helicoid transverse to an applied field is re-entrant in region (II). Region (III) is a half-skyrmion lattice with defects. (**b**) Dependencies of energy densities in all considered modulated phases on the applied magnetic field $h$ ($a = 0.23$) are calculated with respect to the conical phase. The evolution of the half-skyrmion lattice (**c**) for the field pointing opposite to the $z$ axis exhibits transformation into a helicoid. The $-\pi$ SkL (**d**) also turns into a helicoid for the same field direction. For the $+\pi$ SKL (**e**), the field is pointing along the $z$ direction, i.e., along the magnetization in the cell center. All structures are shown with the help of contour plots for the $m_z$-component of the magnetization and correspond to the points in (**b**).

## 3. Phase Diagram of Solutions for $\eta = 0.8$

The magnetic phase diagram (Figure 3a) calculated for $\eta = 0.8$ is drastically different from the typical phase diagram of cubic helimagnets. It is a manifestation of the fact that even a small difference between angular and longitudinal stiffnesses may contribute to much more complicated behavior in the precursor region. The present phase diagram includes pockets with a square half-skyrmion lattice, hexagonal lattice with the magnetization in the center of the cells parallel to the applied magnetic field, and helicoids with propagation transverse to the field. At low fields, a half-skyrmion staggered lattice is the global minimum of the system. At lines E-A and A-C, this lattice undergoes a first-order phase transition into the conical phase and the $+\pi$-skyrmion lattice, correspondingly. At a higher field, the $+\pi$-skyrmion lattice competes with a helicoidal phase with the line B-C being the line of a first-order phase transition between them. In contrast, the $-\pi$-

skyrmion lattice states expected to form a metastable low-temperature phase in chiral cubic helimagnets [16,35] do not exist near magnetic ordering in this model. Critical points of this phase diagram have the following coordinates: A = (0.209, 0.029), B = (0.204, 0.036), D = (0.265, 0), and E = (0.152, 0).

The phase diagram shows that both helicoidal kink-like and skyrmionic precursors may exist.

## 4. Field- and Temperature-Driven Transformation of Modulated States for $\eta = 0.8$

In Figure 3b, the energy densities of all considered modulated phases are plotted with respect to the energy of the conical phase. The snapshots of the contour plots for $m_z$-components of the magnetization in particular points of these curves are shown in panels (c) and (d) of Figure 3. These contour plots provide basic insight into the transformation of different modulated phases in the applied magnetic field.

### 4.1. Transformation of the $-\pi$-Skyrmion Lattice in Applied Magnetic Field

For $\eta = 0.8$, the hexagonal lattice of $-\pi$-skyrmions represents the metastable state with the largest energy density from all skyrmion textures (red curve in Figure 3b). In the applied magnetic field, the energy density of the $-\pi$-skyrmion lattice increases (points $m_1$ and $m_2$), and eventually at some critical magnetic field $h(m_3)$, the skyrmion lattice undergoes the transformation toward the spiral state with the lower energy density (Figure 3d). At the field $h(n_3)$, which is the intersection point of skyrmion and spiral energy densities (red and dark-blue curves in Figure 3b), the first-order phase transition occurs between metastable helical and $-\pi$-skyrmion states. To obtain a numerical solution for the $-\pi$-skyrmion lattice, the temperature $T_k$ of the Monte Carlo annealing must be relatively low (see Section 2.2 on the details of the numerical methods). Otherwise, $-\pi$-skyrmions transform into the state with the lowest energy for $h < h(m_3)$ and even for $h = 0$. For $h < h(n_3)$, $-\pi$-skyrmions turn into the half-skyrmion square lattice (blue curve in Figure 3b); for $h(n_3) < h < h(m_3)$, into the helicoid (dark-blue curve in Figure 3b).

In Figure 3d, the structure of the skyrmion lattice is characterized by the contour plots for the $m_z$ component of the magnetization. The magnetic field is applied down, i.e., along the magnetization in the centers of triangular regions (blue triangles surrounding the main hexagon in Figure 3d, points $m_1$ and $m_2$), and significantly increases their fraction with respect to the parts of the lattice with opposite directions of the magnetization. In the point $m_3$, the lattice loses its stability and elongates into the spiral. In Figure 3d (point $m_3$), the initial moment of the transformation is shown.

### 4.2. Transformation of the $+\pi$-Skyrmion Lattice in Applied Magnetic Field

The $+\pi$-skyrmion lattice (black curve in Figure 3b) is the metastable state in the interval of magnetic fields $0 < h < h(n_1)$. In point $n_1$, the first-order phase transition occurs between half- and $+\pi$-skyrmion lattices. In the interval of fields $h(n_1) < h < h(n_2)$, $+\pi$-skyrmions are the global minimum of the system. In point $n_2$, the helicoids replace the skyrmions by the first-order phase transition. In the phase diagram (Figure 3a), the region of thermodynamical stability of $+\pi$-skyrmions is displayed by the hatching. For $h < h(n_1)$, $+\pi$-skyrmions can be easily transformed into the square lattice of half-skyrmions as shown by the dotted line in Figure 3b. Therefore, the temperature of the Monte Carlo annealing must be sufficiently low.

In the applied magnetic field, the fraction of the skyrmion lattice with the magnetization along the field grows rapidly at the expense of the triangular regions with the opposite magnetization (point $p_1$ in Figure 3d). For the fields $h > h(n_2)$, there are two scenarios for the evolution of this skyrmion lattice: in the first variant, the $+\pi$- skyrmion lattice turns into the helicoid as it was described also for $-\pi$-skyrmions; alternatively, $+\pi$-skyrmions may transform into the homogeneous state.

### 4.3. Transformations of the Half-Skyrmion Lattice

For $\eta < 1$, the half-skyrmion lattice is the global minimum of the system in the interval of magnetic fields $0 < h < h(n_1)$ (blue line in Figure 3b). Additional energy costs to make the magnetization zero along particular directions in the square lattice are lower than for $\eta > 1$. As a result, the region of square lattice lability broadens essentially. For $\eta = 0.8$, the half-skyrmion lattice is thermodynamically stable in the temperature interval $0.152 < a < 0.265$, $h = 0$ (see phase diagram in Figure 3a).

In the applied magnetic field, the relative area of plaquettes in the half-skyrmion lattice magnetized along the field grows at the cost of the oppositely magnetized plaquettes ($h_2$ in Figure 3c). For $h > h(n_1)$, the half-skyrmion lattice may either transform into the more stable $+\pi$-skyrmion lattice (point $h_3$ in Figure 3c) with the subsequent transformation into the helicoid or elongate into the spiral state through intermediate structures shown in Figure 3c, $h_4$. Energy density has a local minimum for such modulated states.

The region of thermodynamical stability of the half-skyrmion lattice is marked in blue in Figure 3a.

### 4.4. Transformation of Helicoids in the Applied Magnetic Field

For definiteness, one-dimensional helical states will be considered to propagate along the $y$-coordinate axis; an applied magnetic field is directed along $z$ (Figure 4a). Rotating magnetization **m** is written in spherical coordinates,

$$\mathbf{m} = m(y)\,(\sin\theta(y), \cos\theta(y), 0), \tag{8}$$

with $\theta(y)$ being the angle of the magnetization with respect to the $z$ axis and $m(y)$—the longitudinal order parameter.

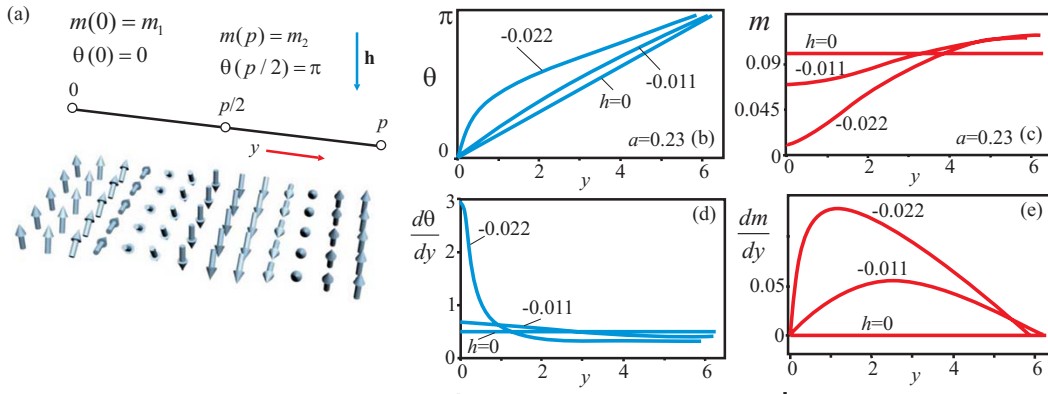

**Figure 4.** (Color online) Solutions for the helicoid (**a**) presented as dependencies $\theta(y)$ (**b**), $d\theta/dy(y)$ (**d**), $m(y)$ (**c**), $dm/dy(y)$ (**e**) demonstrate strong transformation of the helical structure in the applied magnetic field for $a = 0.23$. The longitudinal value of the magnetization along the field gradually increases, whereas opposite to the field, it decreases (see sketch in (**a**) and longitudinal profiles in (**c**)). Angular profiles become more localized (see solutions in (**b**)). In a critical magnetic field $h = -0.024$, the magnetization opposite to the field is equal to zero $m_1(0) = 0$.

The energy density of such a helical state after substituting (8) into Equation (3) can be written as

$$\Phi = m^2\left(\frac{d\theta}{dy}\right)^2 - m^2\frac{d\theta}{dy} + \eta\left(\frac{dm}{dy}\right)^2 + am^2 + m^4 - hm\cos\theta \tag{9}$$

The Euler equations

$$\frac{d^2\theta}{dy^2} + \frac{2}{m}\frac{dm}{dy}\frac{d\theta}{dy} - \frac{1}{m}\frac{dm}{dy} - \frac{h}{2m}\sin\theta = 0,$$

$$\frac{d^2m}{dy^2} - \frac{m}{\eta}\left(\left(\frac{d\theta}{dy}\right)^2 - \frac{d\theta}{dy} + a + 2m^2\right) + \frac{h}{2\eta}\cos\theta = 0 \qquad (10)$$

with boundary conditions

$$\theta(0) = 0,\ \theta(p/2) = \pi,\ m(0) = m_1,\ m(p/2) = m_2 \qquad (11)$$

describe the structure of the helicoid in dependence on the values of the applied magnetic field $h$. $p$ is a period of the helicoid.

In Figure 4b–e, I have plotted the dependencies $m = m(y)$ (c) and $\theta = \theta(y)$ (b) as well as $dm/dy(y)$ (e) and $dm/dy(y)$ (d) in the helicoid for different values of the field. In a zero magnetic field, the magnetization with the constant modulus performs the single-mode rotation around the propagation direction. The longitudinal and angular order parameters are analytically defined as

$$m = \sqrt{\frac{0.25 - a}{2}},\ \theta = \frac{y}{2}. \qquad (12)$$

Increasing magnetic field $\mathbf{h}||z$ destroys the single-mode character of rotation in the helicoid: the magnetic field stretches the value of the magnetization along the field ($m_2$ in Figure 4c) and squeezes it for the opposite direction ($m_1$ in Figure 4c). The angular profiles become strongly localized (blue lines in Figure 4b). Dependencies of derivatives for corresponding order parameters are also highly non-linear (Figure 4d,e): the magnetization vector tries to rotate faster in the parts of the helicoid opposite to the field.

For some critical value of the magnetic field (in Figure 4 for $a = 0.23$, this critical field is 0.024), the value of $m_1(0)$ decreases to zero. In the further increasing magnetic field as a possible solution to Equation (10) and, therefore, a candidate of the helicoid evolution, I considered the one-dimensional spiral state with the following boundary conditions:

$$\theta(0) = 0,\ \theta(p/2) = \theta_0,\ m(0) = m_1,\ m(p/2) = 0. \qquad (13)$$

In Figure 5, the same characteristic features for this spiral state as in Figure 4 are depicted. The length of the magnetization along the field continuously increases, whereas the angle $\theta(p/2)$ decreases.

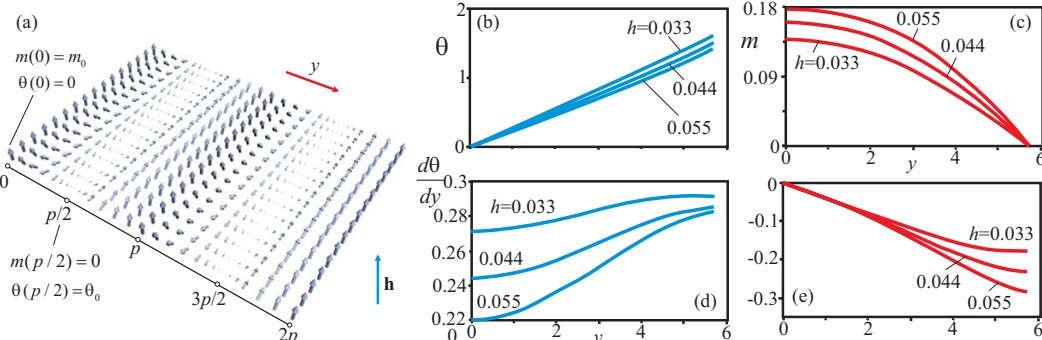

**Figure 5.** (Color online) Solutions for the one-dimensional modulated state with boundary conditions (13) presented as dependences $\theta(y)$ (**b**), $d\theta/dy(y)$ (**d**), $m(y)$ (**c**), $dm/dy(y)$ (**e**). Such a state is considered as a possible scenario for the evolution of a helicoid in a strong magnetic field. In (**a**) the structure of the helical state is presented schematically.

Considered that the helicoid is the global minimum of the system in the range of fields, $h(\mathrm{n}_2) < h < h(\nu_1)$ (Figure 3b). In the point $\mathrm{n}_2$, it replaces, by the first-order phase transition, the $+\pi$-skyrmion lattice. Point $\nu_1$ marks the first-order phase transition with a homogeneous state. For $h > h(\nu_1)$, such a helicoid can still exist but as a metastable solution with the positive energy density. In Figure 3a, the region of the helicoid stability is shown by the light violet color.

## 5. Conclusions

The basic phenomenological model for chiral ferromagnets (Equation (1)) allows to obtain rigorous solutions for 2D skyrmions and analytical solutions for one-dimensional helical and conical states in the whole range of the control parameters—the reduced values of temperature $a$, and magnitude of the applied magnetic field $h$. The properties of the chiral modulations reveal a noticeable similarity with characteristic peculiarities of cubic helimagnets near the ordering temperatures known as "precursor states" and "A-phase anomalies". This allows to suggest that the softening of the magnetization magnitude is the basic physical mechanism underlying anomalous properties of "precursor states" in chiral magnets. As the energy differences between different modulated phases in the high-temperature region are very small, additional energy contributions result in changes of relative phase stabilities and may cause the drastic modification of phase diagrams: cubic and exchange anisotropies are known to stabilize $-\pi$-skyrmions not only in the particular interval of the magnetic field near $T_c$ but also close to zero temperatures. Additional softening of the magnetization modulus considered within the non-Heisenberg model allows to favor a rich variety of modulated states with zero-modulus lines and points substituting the notion of defects. Then, half-skyrmions and spirals with $m(p/2) = 0$ may also occupy vast stability regions at the phase diagrams. This effect comes into play already for minor differences between stiffnesses of longitudinal and angular order parameters, thus making the precursor region more intricate. Remarkably, the concept of skyrmionic textures in chiral magnetic systems can be extended to continuum models for glass-forming liquids. These models describe the frustrated tiling of space by the incompatible locally preferred clusters of a molecular liquid within a generalized elastic theory. The field theory for the local order-parameter includes antisymmetric couplings derived from the decurving of ideal template units into flat space. As a qualitatively new feature, a model with a softened modulus of the local intensity of the order parameter was proposed in Ref. [37]. The corresponding classical field theory allows the stabilization of skyrmionic localized states and extended textures. The notion of a glassy structure as an entangled network of defect lines is replaced by the complex geometry of an elastic and frustrated continuum that can display both "rotation" or twisting and longitudinal suppression of the ideal local order. The skyrmions in the simplest version of the frustration models are close but soft relatives of the hedgehog solutions in Skyrme's original SU(2) symmetric model for nucleons [38]. It is argued that stable skyrmions are formed at elevated temperatures in molecular liquids and that their condensation into frustrated textures underlies the stability of supercooled and glassy states.

**Funding:** This research was funded by JSPS grant number 21K03406 for A.O.L.

**Institutional Review Board Statement:** Not applicable

**Informed Consent Statement:** Not applicable

**Data Availability Statement:** The data presented in this study are available on request from the corresponding author.

**Acknowledgments:** A.O.L. acknowledges JSPS Grant-in-Aid (C) 21K03406.

**Conflicts of Interest:** The author declares no conflict of interest.

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
