# Peer review of "Chiral Modulations in Non-Heisenberg Models of Non-Centrosymmetric Magnets Near the Ordering Temperatures"

_2673-8724, doi:10.3390/magnetism4020007_

Round 1

Reviewer 1 Report

Comments and Suggestions for Authors

The paper is devoted to the skyrmion lattice type of structures in the Dzyaloshinskii model. In the present work the author extends his previous work [13] on the generalized gradient energy functional. The model differs from the Heisenberg-like exchange interaction model by an "anisotropic" stiffness of the absolute value of the magnetization. The author claims that small changes in the stiffness of the absolute value of the magnetization strongly affect the phase diagram.

The paper contains interesting new results and may be published in this journal after the following questions are addressed:

1) The author has mentioned three affiliations, but only two of them are attributed to him. What is the role of the IFW Dresden?

2) Fig. 1a. This figure is similar to Fig1a [13], but the turning points are different. What is the reason for these differences? Different functionals? Please comment on the reason for the difference in the paper.

3) Fig1b. - The labelling of the x-axis should be clearer (same font and position as in Fig1a).

4) The transformation of Eq.(2) is different from [13]. Please comment.

5) Below Eq.(4) the author claims that the interaction between the skyrmions (attraction, repulsion) follows from the Euler equations given by Eqs.(4). This statement is important and plays one of the central roles in the paper. However, it is unclear to a reader how to derive the statement. Please comment on what terms determine the interaction. 

6) The phase diagram contains a region of attraction between the skyrmions and the region of skyrmion confinement. Please add the definition of skyrmion confinement.

7) The conclusion gives the impression that the main application of the model is to metallic glasses. I suggest that this model should be discussed in detail and the relationship of the results to this model should be shown. The author may devote a separate section to this subject.

Reviewer 2 Report

Comments and Suggestions for Authors

In this paper, A. Leonov investigated the skyrmion and spiral states under modified magnetic energy, including a fourth power term and an anisotropic stiffness term. The phase diagram of the skyrmion and spiral states are solved via a Monte Carlo algorithm, as function of effective magnetic field and effective temperature. These investigations reveal the important role of nonlinear and anisotropic interactions in magnetic states. Therefore, this paper can be accepted upon proper modifications as follows:

1.     The fourth power term in Eq. (1) is introduced without any reference. While its implication as an effective temperature is reasonable, deeper discussions on this term are suggested.

2.     The meaning of chiral modulation in the manuscript title deserves better clarification.

3.     Several section titles are not well organized, such as in section 2.3 and section 4.

Reviewer 3 Report

Comments and Suggestions for Authors

Referee’s report on “Chiral modulations in non-Heisenberg models near the ordering temperatures”.

In this manuscript, the author used Monte Carlo simulations to study the spin model described by Eq. 3. A rich field-temperature phase diagram is obtained for \eta=0.8 (Eq. 7), which they attribute to the softness of the longitudinal order parameter. They described the behavior of each phase in detail. In general, I find these numerical results interesting, as they might guide the controlled realization of some phases. As a result, I recommend publishing this work. However, I find some parts hard to follow for general readership. My comments are provided below for them to improve this manuscript.

Lines 29-30: The observation of a skyrmion lattice in MnSi is not a proposal. It is an unambiguous experimental realization.

Lines 35-36: Some of the references cited do not support the author’s claim that the phase boundaries are changed. The authors should use a more precise language.

Line 36: “Consequently, …” I do not see any relevance of this sentence to the sentences above. Why using consequently?

Lines 40-50: Several different types of skyrmions, e.g. half- and +/-\pi skyrmions are mentioned without definition. This, including their distinction from the conventional skyrmions, should be addressed here before describing them in Fig,2 and their responses to field and temperature in the main text.

Lines 127-128: How was the initial configuration chosen? What does it mean by choosing it appropriately? Can they start with a random configuration? 

Line 307: What is a metastable solution in numerical simulations? Is this phase not favored if we let the system relax longer?

Lines 330-344: The discussion about glass-forming liquids is out of context. I suggest removing it.

Comments on the Quality of English Language

NA

Reviewer 4 Report

Comments and Suggestions for Authors

This is good research, I would say it is a continuation of the latest contributions from this author's group, cited here as well. The following aspects require clarification and/or improvements:

1) The use of Monte Carlo language is misleading, that is, is Monte Carlo dynamics really a dynamic process, or does the author simply call the Monte Carlo steps time steps? The Metropolis algorithm appears to implement an ergodic and detailed-balance process for a typical thermalization of a non-dynamical Markovian process. It is necessary to clarify the relationship between the Monte Carlo steps (and what to change between them) and temporal evolution. This could be a crucial point within the scope of this type of investigation.

2) The softening of the magnetization modulus, which is considered here within a non-Heisenberg model, must be put into perspective from an experimental realization. Please provide more references or ideas on how that could be achieved in a material. In reality, there are, for example, new iron oxides that could effectively present varying local magnetizations due to fast FM/AFM switching properties promoted by oxygen migration (which could be driven by another field). For those, within a midfield scope, something like this might be plausible. Could this softening of magnetization be inevitably related to the appearance of multiferroic effects?

3) The conclusion section is overlong/inconclusive, especially in light of the author's previous publications. What are the conclusions of these manuscripts alone?

It shouldn't be difficult for the author to improve the article by looking at those comments so that it is really clear to the readers of the journal.
